# Witten index of BMN matrix quantum mechanics

Chi-Ming Chang

*Yau Mathematical Sciences Center (YMSC), Tsinghua University, Beijing 100084, China*

*Beijing Institute of Mathematical Sciences and Applications (BIMSA)*
*Beijing 101408, China*

cmchang@tsinghua.edu.cn

## Abstract

We compute the Witten index of the Berenstein-Maldacena-Nastase matrix quantum mechanics, which counts the number of ground states as well as the difference between the numbers of bosonic and fermionic BPS states with nonzero spins. The Witten index sets a lower bound on the entropy, which exhibits an $N^2$ growth that predicts the existence of BPS black holes in M-theory, asymptotic to the plane wave geometry. We also discuss a relation between the Witten index in the infinite $N$ limit and the superconformal index of the Aharony-Bergman-Jafferis-Maldacena theory.

# 1   Introduction

The Berenstein-Maldacena-Nastase (BMN) matrix quantum mechanics [1] is a supersymmetric gauged quantum mechanics with sixteen supercharges. It is a mass deformation of the Banks-Fischler-Shenker-Susskind (BFSS) matrix quantum mechanics [2], and converges to the BFSS matrix quantum mechanics when the dimensionless effective gauge coupling $g$ (normalized by the mass scale) approaches infinity.

The BMN matrix quantum mechanics participates in a very rich array of dualities. First, by the BFSS conjecture [2], the BMN matrix quantum mechanics is dual to M-theory on the uncompactified plane wave background [3], when the rank $N$ of the matrices approaches infinite with the gauge coupling $g$ fixed. A stronger version of the conjecture states that, at finite $N$ and $g$, the BMN matrix quantum mechanics is dual to the Discrete Lightcone Quantization (DLCQ) of M-theory on the plane wave background with $N$ unit of light cone momentum [4–6]. Finally, in the 't Hooft limit ($\lambda = g^2 N$ fixed and $N \to \infty$), the BMN matrix quantum mechanics is dual to M-theory on asymptotically plane wave backgrounds [7,8], for

instance, at large 't Hooft coupling ($\lambda \gg 1$), the vacuum states are dual to the Lin-Lunin-Maldacena geometries [9, 10], and the deconfined phase at high temperature is dual to black holes [11].

Witten index is a very powerful tool in the study of supersymmetric quantum mechanics [12]. It is defined as a trace over the Hilbert space, weighted by the Boltzmann factor and an additional insertion of the fermion parity operator. The trace is equal to the number of bosonic ground states minus the number of fermionic ground states because there are equal numbers of bosonic and fermionic states at each non-zero energy level of the Hamiltonian due to supersymmetry. Hence, Witten index is independent of the temperature and any deformation of the Hamiltonian as long as the deformation is gentle enough that does not change the Hilbert space. When the theory has global symmetries that commute with the Hamiltonian, the Witten index could be further refined by incorporating flavor fugacities and receiving contributions from not only ground states but also BPS states with non-zero symmetry charges.

The Witten index of the BMN matrix quantum mechanics can be computed in the weak coupling (large mass) limit due to the protection from supersymmetry. In this limit, the theory divides into superselection sectors associated with each supersymmetric vacuum, that contains a set of free bosonic and fermionic harmonic oscillators [13]. The spectrum in each superselection sector is thereby generated by acting on the supersymmetric vacuum state with arbitrary numbers of creation operators of the oscillators. The result could be assembled into a matrix integral formula for the Witten index (see Section 2.3), that is similar to the formula for the superconformal index of the $\mathcal{N} = 4$ super-Yang-Mills (SYM) theory [14]. The result is different from the Witten index of the BFSS matrix quantum mechanics [15, 16], as the Hilbert space at infinite coupling (zero mass) is different from the one at finite coupling due to the flat directions in the classical potential.

The Witten index restricted in different superselection sectors exhibits very different behaviors in the large $N$ limit. We will focus on the sectors associated with the trivial vacuum (the maximally rediculble vacuum) and the irreducible vacuum. In the trivial vacuum sector, the matrix integral can be evaluated directly as a power series of the fugacities at finite $N$, or using a saddle point approximation in the large $N$ and small chemical potentials limit. Both results exhibit an $N^2$ growth in the entropy. This strongly indicates the existence of BPS black holes, analogous to the non-BPS black holes [11], in the bulk dual of the BMN matrix quantum mechanics (see Section 2.4). In the irreducible vacuum sector, BMN matrix quantum mechanics describes a single M2-brane with $N$ units of lightcone momentum along the M-theory circle [1]. On the other hand, a system of coinciding M2-branes is described by the Aharony-Bergman-Jafferis-Maldacena (ABJM) theory [17]. We find that the Witten index in the irreducible vacuum sector in the large $N$ limit is equal to the superconformal

index of the $U(1)_1 \times U(1)_{-1}$ ABJM theory [18] up to a divergent factor (see Section 2.5).

# 2 BMN index

## 2.1 BMN matrix quantum mechanics

The Hamiltonian of the BMN matrix quantum mechanics is [19]

$$
\begin{aligned}
H = &R \operatorname{Tr} \left[ \frac{1}{2} \sum_{I=1}^{9} (P^I)^2 - \frac{1}{4\ell_P^6} \sum_{I,J=1}^{9} [X^I, X^J]^2 - \frac{1}{2\ell_P^3} \Psi^T \gamma^I [X^I, \Psi] \right] \\
&+ \frac{R}{2} \operatorname{Tr} \left[ \left(\frac{\mu}{3R}\right)^2 \sum_{i=1}^{3} (X^i)^2 + \left(\frac{\mu}{6R}\right)^2 \sum_{m=4}^{9} (X^m)^2 + i\frac{\mu}{4R} \Psi^T \gamma^{123} \Psi + i\frac{2\mu}{3R\ell_P^3} \epsilon_{ijk} X^i X^j X^k \right],
\end{aligned}
\tag{2.1}
$$

where $X^I$ for $I = 1, \cdots, 9$ are nine $N \times N$ bosonic matrices, $\Psi$ with a spinor index supressed denotes sixteen $N \times N$ fermionic matrices, and $P^I$ are the conjugate momenta of $X^I$. $X^i$ and $X^m$ for $i = 1, 2, 3$ and $m = 4, \cdots, 9$ are the first three and last six components of $X^I$. We will focus on the $U(N)$ gauge group, and the gauge symmetry acts on the Hermitian matrices $X^I$, $\Psi$, and $P^I$ as conjugation by unitary matrices. The parameters $R$, $\ell_P$ and $1/\mu$ have dimension of length. $\ell_P$ could be obsorbed by field and parameter redefinitions: $X^I = \ell_P \widetilde{X}^I$ and $R = \ell_P^2 \widetilde{R}$, so that $\widetilde{X}^I$ are dimensionless and $1/\widetilde{R}$ has dimension of length. Therefore, there is only a single dimensionless coupling

$$
g^2 = \frac{R^3}{\mu^3 \ell_P^6} \, .
\tag{2.2}
$$

At infinite coupling $g \to \infty$ ($\mu = 0$), the BMN matrix quantum mechanics becomes the Banks-Fischler-Shenker-Susskind (BFSS) matrix quantum mechanics [2], with the Hamiltonian given by the first line of (2.1) that preserves SO(9) symmetry. $X^I$, $P^I$ are in the vector representation and $\Psi$ is in the spinor representation. The potential terms on the second line of (2.1) breaks the SO(9) to SO(3) $\times$ SO(6).

## 2.2 Witten index

The BMN matrix quantum mechanics has SU(2|4) supersymmetry, which contains the SO(3)$\times$SO(6) symmtry as the maximal bosonic subgroup. The SU(2|4) supersymmetry contains sixteen supercharges $Q_\alpha^m$ and $(Q_\alpha^m)^\dagger$, where $\alpha = \pm$ is the spinor index of SU(2) $\cong$ SO(3). The supercharges have the anti-commutator [13]

$$
\{Q_\alpha^m, (Q_\beta^n)^\dagger\} = 2\delta_n^m \delta_\beta^\alpha H + \frac{\mu}{3} \epsilon_{ijk} (\sigma^k)_\alpha^\beta \delta_n^m M^{ij} - \frac{2\mu}{3} \delta_\beta^\alpha R_n^m \, ,
\tag{2.3}
$$

where $M^{ij}$ and $R_n^m$ are the rotation generators of SO(3) and SU(4), respectively. Let us pick a supercharge $Q := Q_-^4$. The anti-commutator of $Q$ with its Hermitian conjugate $Q^\dagger$ is

$$2\Delta := \{Q, Q^\dagger\} = 2H - \frac{2\mu}{3}M^{12} - \frac{2\mu}{3}R_4^4 = 2H - \frac{2\mu}{3}M^{12} - \frac{\mu}{3}(M^{45} + M^{67} + M^{89}), \quad (2.4)$$

where we expand $R_4^4$ in terms of the Cartan generators $M^{45}$, $M^{67}$, $M^{89}$ of SO(6), which correspond to the rotations along the three orthogonal two-planes in $\mathbb{R}^6$. It is convenient to assemble the 5 Cartan generators of SU(2|4) into a vector as

$$\left(\frac{12H}{\mu}, 4M^{12}, 2M^{45}, 2M^{67}, 2M^{89}\right). \quad (2.5)$$

The supercharge $Q$ has the Cartan charges $(1, -2, 1, 1, 1)$.

Let us consider the thermal partition function

$$Z = \text{Tr}\,\Omega, \quad \Omega := e^{-\beta\Delta - 2\omega M^{12} - \Delta_1 M^{45} - \Delta_2 M^{67} - \Delta_3 M^{89}}. \quad (2.6)$$

The Boltzmann factor $\Omega$ anti-commutes with the supercharge $Q$ if the chemical potentials satisfy the linear relation

$$\Delta_1 + \Delta_2 + \Delta_3 - 2\omega = 2\pi i \quad \text{mod } 4\pi i. \quad (2.7)$$

The Witten index is defined by

$$\mathcal{I} = Z\big|_{(2.7)} = \text{Tr}\left[(-1)^F e^{-\beta\Delta - \Delta_1(M^{12}+M^{45}) - \Delta_2(M^{12}+M^{67}) - \Delta_3(M^{12}+M^{89})}\right], \quad (2.8)$$

where we identify $2M^{12}$ with the fermion number $F$ beause the fermion fields $\Psi$'s have half-integer $M^{12}$-eigenvalues. The Witten index $\mathcal{I}$ is independent of the inverse temperature $\beta$, because the states with nonzero $\Delta$-eigenvalues are all paired up by the action of the super-charge $Q$ and their contributions to the Witten index cancel. Since all the explicit coupling constant $g$ dependence is inside the Hamiltonian, the Witten index is also independent of $g$ away from the infinite coupling ($g = \infty$) point where the dimension of the Hilbert space becomes uncountable due to continuums in the spectrum.

## 2.3 Computation at weak coupling

Since the Witten index is independent of the coupling constant $g$, it can be computed for BMN matrix quantum mechanics in the weak coupling (small $g$) limit, or equivalently, in the large $\mu$ limit. We will begin by reviewing the computation of the weak coupling spectrum in [13], and then use the results to compute the Witten index.

In the large $\mu$ limit, the bosonic potential becomes very steep, and we could expand the Hamiltonian about the minima of the potential (classical supersymmetric vacua). The bosonic potential can be written in a manifestly positive form as

$$V = \frac{R}{2}\text{Tr}\left[\left(\frac{\mu}{3R}X^i + \ell_P^{-3}i\epsilon^{ijk}X^jX^k\right)^2 + \frac{1}{2\ell_P^2}(i[X^m, X^n])^2 + \frac{1}{\ell_P^2}(i[X^m, X^i])^2 + \left(\frac{\mu}{6R}\right)^2(X^m)^2\right].$$
(2.9)

The classical supersymmetric vacua are given by [1]

$$X^m = 0, \quad X^i = \frac{\mu\ell_P^3}{3R}J^i,$$
(2.10)

where $J^i$ is a $N$-dimensional matrix representation of SU(2), i.e. $[J^i, J^j] = i\epsilon^{ijk}J^k$. Any $N$-dimensional representation of SU(2) can be written as a direct sum of irreducible representations, and is labeled by the partition of the integer $N$,

$$N = \sum_{k=1}^{K} n_k N_k.$$
(2.11)

More explicitly, the $(k, l)$-th block of $J^i$ is given by

$$J_{kl}^i = \delta_{kl}I_{n_l} \otimes J_l^i,$$
(2.12)

where $J_l^i$ is the $N_l$-dimensional irreducible representation, and $I_{n_l}$ is a $n_l \times n_l$ identity matrix. In the trivial vacuum ($K = 1$, $n_1 = N$, and $N_1 = 1$), the U($N$) gauge symmetry is preserved. In the nontrivial vacua, the U($N$) gauge symmetry is broken to

$$\text{U}(n_1) \times \cdots \times \text{U}(n_K).$$
(2.13)

The classical vacua correspond to degenerate ground states, which are protected quantum mechanically and have vanishing energy at finite coupling $g > 0$ [20–22].

The expansion about the classical vacua was studied in [13]. In the small $g$ limit, the Hamiltonian reduces to a free quadratic term plus higher order interaction terms, which are suppressed by powers of $g$ relative to the quadratic term. Hence, the theory divides into superselection sectors associated with each classical vacuum and labeled by the partition (2.11) of $N$. Each sector contains a single ground state, and excited states given by acting creation operators of a set of bosonic and fermionic harmonic oscillators. In the $(k, l)$-th block, the bosonic oscillators are

$$(\alpha_{kl})_{jm}, \quad (\beta_{kl})_{jm}, \quad (x_{kl}^a)_{jm},$$
(2.14)

and fermionic oscillators are

$$(\chi_{kl}^I)_{jm}, \quad (\eta_{I,kl})_{jm},$$
(2.15)

which are all $n_k \times n_l$ matrices, and transform in the bi-fundamental representation of $\mathrm{U}(n_k) \times \mathrm{U}(n_l)$. The ranges of $j$ for each oscillator are given in Table 2 in [13]. For example, the matrix $X_{kl}^a$ has the expansion

$$X_{kl}^a = \sum_{j=\frac{1}{2}|N_k-N_l|}^{\frac{1}{2}(N_k+N_l)-1} \sum_{m=-j}^{j} (x_{kl}^a)_{jm} \otimes Y_{jm}^{N_k N_l} , \tag{2.16}$$

where $Y_{jm}^{N_k N_l}$ is a $N_k \times N_l$ matrix as a spin-$j$ representation in the tensor product of the spin-$\left(\frac{N_k-1}{2}\right)$ and spin-$\left(\frac{N_l-1}{2}\right)$ representations.

We focus on BPS excited states, which are given by acting creation operators of the BPS letters, the oscillators satisfying the BPS condition $\Delta = 0$. The BPS letters are listed in Table 1.

| letter | charges | index |
|--------|---------|-------|
| $\beta$ | $(4j, 4j, 0, 0, 0)$ | $e^{-2j\omega}$ |
| | $(4j+2, 4j, 2, 0, 0)$ | $e^{-2j\omega-\Delta_1}$ |
| $x$ | $(4j+2, 4j, 0, 2, 0)$ | $e^{-2j\omega-\Delta_2}$ |
| | $(4j+2, 4j, 0, 0, 2)$ | $e^{-2j\omega-\Delta_3}$ |
| $\chi$ | $(4j+3, 4j, 1, 1, 1)$ | $e^{-2j\omega-\frac{\Delta_1}{2}-\frac{\Delta_2}{2}-\frac{\Delta_3}{2}}$ |
| | $(4j+1, 4j, 1, 1, -1)$ | $e^{-2j\omega-\frac{\Delta_1}{2}-\frac{\Delta_2}{2}+\frac{\Delta_3}{2}}$ |
| $\eta$ | $(4j+1, 4j, 1, -1, 1)$ | $e^{-2j\omega-\frac{\Delta_1}{2}+\frac{\Delta_2}{2}-\frac{\Delta_3}{2}}$ |
| | $(4j+1, 4j, -1, 1, 1)$ | $e^{-2j\omega+\frac{\Delta_1}{2}-\frac{\Delta_2}{2}-\frac{\Delta_3}{2}}$ |

Table 1: The BPS letters and their charges (in the vector form (2.5)) and indices.

The single-letter partition function for the BPS letters is

$$
\begin{aligned}
z_{kl}(\omega, \Delta_i) = & \sum_{j=\frac{1}{2}|N_k-N_l|}^{\frac{1}{2}(N_k+N_l)-1} e^{-2j\omega}\left(e^{-\Delta_1} + e^{-\Delta_2} + e^{-\Delta_3}\right) \\
& + \sum_{j=\frac{1}{2}|N_k-N_l|+1}^{\frac{1}{2}(N_k+N_l)} e^{-2j\omega} + \sum_{j=\frac{1}{2}|N_k-N_l|-\frac{1}{2}}^{\frac{1}{2}(N_k+N_l)-\frac{3}{2}} e^{-2j\omega-\frac{\Delta_1}{2}-\frac{\Delta_2}{2}-\frac{\Delta_3}{2}} \\
& + \sum_{j=\frac{1}{2}|N_k-N_l|+\frac{1}{2}}^{\frac{1}{2}(N_k+N_l)-\frac{1}{2}} e^{-2j\omega}\left(e^{-\frac{\Delta_1}{2}-\frac{\Delta_2}{2}+\frac{\Delta_3}{2}} + e^{-\frac{\Delta_1}{2}+\frac{\Delta_2}{2}-\frac{\Delta_3}{2}} + e^{\frac{\Delta_1}{2}-\frac{\Delta_2}{2}-\frac{\Delta_3}{2}}\right).
\end{aligned}
\tag{2.17}
$$

Imposing the relation (2.7) among the chemical potentials, we find the single-letter index

$$
\begin{aligned}
\iota_{kl}(\Delta_i) &= z_{kl}(\omega, \Delta_i)\big|_{(2.7)} \\
&= \sum_{j=\frac{1}{2}|N_k-N_l|}^{\frac{1}{2}(N_k+N_l)-1} (-1)^{2j+1} e^{-j(\Delta_1+\Delta_2+\Delta_3)}(1-e^{-\Delta_1})(1-e^{-\Delta_2})(1-e^{-\Delta_3}) + \delta_{N_k,N_l} \,.
\end{aligned}
\tag{2.18}
$$

The multi-letter index is given by the plethystic exponential of the single-letter index $\iota_{kl}(\Delta_i)$. The Witten index in the superselection sector is given by further imposing the gauge invariance under the unbroken subgroup (2.13) by matrix integrals,

$$
\mathcal{I}^{\mathrm{BMN}}_{n_i;N_i} = \int \prod_{k=1}^{K} [dU_k] \exp\left[ \sum_{m=1}^{\infty} \sum_{k,l=1}^{K} \frac{1}{m} \iota_{kl}(m\Delta_i) \mathrm{Tr}\, U_k^{\dagger m} \mathrm{Tr}\, U_l^m \right] \,,
\tag{2.19}
$$

where $U_k$ is a $n_k \times n_k$ unitary matrix. The total Witten index is given by

$$
\mathcal{I}^{\mathrm{BMN}} = \sum_{n_i,N_i \in \mathbb{Z}_{>0},\,(2.11)} \mathcal{I}^{\mathrm{BMN}}_{n_i;N_i}
\tag{2.20}
$$

## 2.4  Trivial vacuum sector

Let us consider the Witten index in the trivial vacuum sector given by $K = 1$, $n_1 = N$, $N_1 = 1$. The integral formula (2.19) reduces to

$$
\mathcal{I}^{\mathrm{BMN}}_{N;1} = \int [dU] \exp\left\{ \sum_{m=1}^{\infty} \frac{1-(1-e^{-m\Delta_1})(1-e^{-m\Delta_2})(1-e^{-m\Delta_3})}{m} \mathrm{Tr}\, U^{\dagger m} \mathrm{Tr}\, U^m \right\} \,.
\tag{2.21}
$$

The same index $\mathcal{I}^{\mathrm{BMN}}_{N;1}$ was studied in [23, 24] but with the interpretation as the superconformal index of the $\mathcal{N} = 4$ SYM truncated to the BMN sector, that contains only fundamental fields invariant under the chiral $\mathrm{SU}(2)_R$ rotation. It was argued in [24], following a similar analysis in [25, 26], that in the large $N$ limit with small and fixed $\Delta_i$, the matrix integral (2.21) has a saddle point with the eigenvalue distribution

$$
\rho(\alpha) = \frac{3}{4\pi^3}(\pi^2 - \alpha^2) \quad \text{for} \quad \alpha \in (-\pi, \pi) \,.
\tag{2.22}
$$

The saddle point contributes to the index as

$$
\log \mathcal{I}^{\mathrm{BMN}}_{N;1} = -\frac{3N^2}{2\pi^2} \Delta_1 \Delta_2 \Delta_3 \,.
\tag{2.23}
$$

Let us consider the Legendre transform of $\log \mathcal{I}_{N;1}^{\mathrm{BMN}}$ given by the extremization

$$
\begin{aligned}
& S_{N;1}^{\mathrm{BMN}}(M^{12}, M^{45}, M^{67}, M^{89}) \\
& = \underset{\Delta_i}{\mathrm{ext}} \left[ \log \mathcal{I}_{N;1}^{\mathrm{BMN}} + \Delta_1(M^{12} + M^{45}) + \Delta_2(M^{12} + M^{67}) + \Delta_3(M^{12} + M^{89}) \right], \\
& = 2\pi \sqrt{\frac{2(M^{12} + M^{45})(M^{12} + M^{67})(M^{12} + M^{89})}{3N^2}},
\end{aligned} \tag{2.24}
$$

that is valid in the large $N$ limit with $\epsilon_i := (M^{12} + M^{2i+2,2i+3})/N^2$ (for $i = 1, 2, 3$) held fixed and small $\epsilon_i \ll 1$. The function $S_{N;1}^{\mathrm{BMN}}$ sets a lower bound on the entropy of the BPS states; however, for simplicity, we will also refer to $S_{N;1}^{\mathrm{BMN}}$ as the entropy. We find that the entropy $S_{N;1}^{\mathrm{BMN}} \sim \sqrt{\epsilon_1 \epsilon_2 \epsilon_3} N^2$ exhibits $N^2$ scaling, which implies that the theory is in a deconfined phase and should correspond to a BPS black hole in the bulk dual.

A very subtle point of the above analysis is that the contribution from the non-trivial saddle (2.22) is negative (2.23), which is smaller than the contribution from the trivial saddle, corresponding to a confined phase, with a constant eigenvalue distribution

$$
\rho(\alpha) = \frac{1}{2\pi}. \tag{2.25}
$$

However, the entropy (2.24) contributed from this saddle is of order $N^2$, much larger than the entropy from the trivial saddle. Hence, the deconfined phase is subdominant to the confined phase in the grand canonical ensemble, but dominant in the microcanonical ensemble. The deconfined phase has a negative specific heat $C$ (or more precisely the susceptibility) given by[1]

$$
C = \Delta^2 \frac{d^2 \log \mathcal{I}_{N;1}^{\mathrm{BMN}}}{d\Delta^2} = -\frac{9N^2}{\pi^2} \Delta^3 < 0, \tag{2.26}
$$

where we specialize chemical potentials as $\Delta := \Delta_1 = \Delta_2 = \Delta_3$. These behaviors are similar to the small black holes in AdS$_5$ [26], which we review in Appendix A.

An important question is whether the $N^2$ scaling of the entropy persists at finite $\epsilon_i$, and if so, whether the deconfined phase also dominates in the grand canonical ensemble. We provide positive evidence for the $N^2$ scaling from evaluating the BMN index (2.21) as an expansion of the fugacities to high powers. In the microcanonical ensemble, the entropy is defined as the log of the coefficients in the expansion of the index

$$
S_{\mathrm{BMN}}(j) = \log |d_j^{\mathrm{BMN}}|, \quad \mathcal{I}_{N;1}^{\mathrm{BMN}} = \sum_j d_j^{\mathrm{BMN}} t^j, \quad t^2 = e^{-\Delta_1} = e^{-\Delta_2} = e^{-\Delta_3}, \tag{2.27}
$$

where the quantum number $j$ is the following linear combination of angular momenta

$$
j = 6M^{12} + 2(M^{45} + M^{67} + M^{89}). \tag{2.28}
$$

---

[1]I thank Sunjin Choi for a discussion on this point.

For inspecting the the growth of the degeneracy $d_j^{\text{BMN}}$ in the large $N$ limit with $j/N^2$ fixed, we define

$$s_{\text{BMN}}\left(j/N^2\right) = N^{-2}\log|d_j^{\text{BMN}}|. \tag{2.29}$$

Let us set $j = N^2$,[2] and plot $\log|d_j^{\text{BMN}}|$ and $s_{\text{BMN}}$ against $N^2$ in Figure 1. The BMN index looks to achieve convergence when $N \gtrsim 6$, as

$$\log|d_j^{\text{BMN}}| \sim 0.21 \times N^2. \tag{2.30}$$

For comparison, we perform the same analysis for the superconformal index of the $\mathcal{N} = 4$ SYM, which is given by the matrix integral [14]

$$\mathcal{I}_{\mathcal{N}=4} = \int [dU] \exp\left\{ \sum_{m=1}^{\infty} \frac{1 - \frac{(1-e^{-m\Delta_1})(1-e^{-m\Delta_2})(1-e^{-m\Delta_3})}{(1-e^{-m\omega_1})(1-e^{-m\omega_2})}}{m} \text{Tr}\, U^{\dagger m}\text{Tr}\, U^m \right\}, \tag{2.31}$$

with the constraint $\Delta_1 + \Delta_2 + \Delta_3 - \omega_1 - \omega_2 = 2\pi i$. We expand the index as

$$\mathcal{I}_{\mathcal{N}=4} = \sum_j d_j^{\mathcal{N}=4} t^j, \quad t^2 = e^{-\Delta_1} = e^{-\Delta_2} = e^{-\Delta_3}, \quad t^3 = e^{-\omega_1} = e^{-\omega_2}. \tag{2.32}$$

Again, we consider the large $N$ limit with $j/N^2$ fixed and define

$$s_{\mathcal{N}=4}(j/N^2) = N^{-2}\log|d_j^{\mathcal{N}=4}|. \tag{2.33}$$

The entropy $s_{\mathcal{N}=4}(j/N^2)$ can be computed exactly in the grand canonical ensemble by a saddle point approximation [27–29]. When $j = N^2$, we have $s_{\mathcal{N}=4}(1) = 0.357$. The plots of $\log|d_j^{\mathcal{N}=4}|$ and $s_{\mathcal{N}=4}$ against $N^2$ are given in Figure 1. We can see that $s_{\mathcal{N}=4}$ converges to the asymptotic value already at $N \sim 5$.

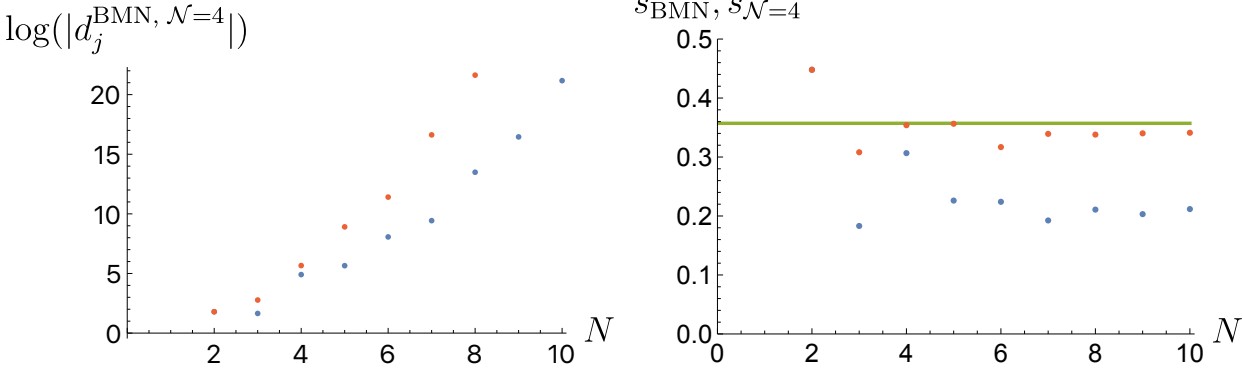

Figure 1: The $\log(|d_j^{\text{BMN}}|)$, $s_{\text{BMN}}$ (blue) and $\log(|d_j^{\mathcal{N}=4}|)$, $s_{\mathcal{N}=4}$ (orange) at $j = N^2$ with increasing $N$. The green line is the asymptotic value for $s_{\mathcal{N}=4}$.

---

[2]There is a small subtlety that the states in the BMN sector all have even $j$. Therefore, when $N$ is odd, we consider the average $\frac{1}{2}(\log|d_{j-1}^{\text{BMN}}| + \log|d_{j+1}^{\text{BMN}}|)$ for $j = N^2$.

## 2.5 Irreducible vacuum sector

Let us consider the Witten index in the sector given by $K = 1$, $n_1 = 1$, $N_1 = N$ corresponding to an irreducible representation of SU(2). The integral formula (2.19) reduces to

$$\mathcal{I}_{1;N}^{\mathrm{BMN}} = \exp\left\{\sum_{m=1}^{\infty} \frac{1}{m}\left[1 - \frac{(1 - e^{-m\Delta_1})(1 - e^{-m\Delta_2})(1 - e^{-m\Delta_3})(1 - e^{-mN(\Delta_1+\Delta_2+\Delta_3)})}{1 - e^{-m(\Delta_1+\Delta_2+\Delta_3)}}\right]\right\}.$$
(2.34)

In this sector, the BMN matrix model describes a single M2-brane carrying $N$ units of lightcone momentum along the M-theory circle and wrapping an $\mathrm{S}^2$ in the plane wave geometry [1]. On the other hand, the ABJM theory with $\mathrm{U}(1)_1 \times \mathrm{U}(1)_{-1}$ gauge group in radial quantization also describes a single M2-brane wrapping an $\mathrm{S}^2$. Hence, we expect some relations between the BMN index and the superconformal index of ABJM theory [18].

In Appendix B, we review the $\mathcal{N} = 8$ ABJM index. It depends on five chemical potentials associated with one spin and four R-charges of the superconformal symmetry OSp(4|8), and with one linear between the chemical potentials. However, only the subgroup SU(2|4) $\subset$ OSp(4|8) is manifest in the BMN matrix quantum mechanics, whose Witten index depends on four chemical potentials with one linear relation. Hence, to compare the BMN index and the ABJM index, one needs to first figure out a relation between the chemical potentials.

**Relation between the chemical potentials**

Under the embedding SU(2|4) $\hookrightarrow$ OSp(4|8), the supercharges in SU(2|4) are related to the supercharges in OSp(8|4) by

$$Q_\alpha^n = \sqrt{\frac{\mu}{3}}(\mathbf{Q}_\alpha^{2n-1} + i\mathbf{Q}_\alpha^{2n}), \quad (Q_\alpha^n)^\dagger = \sqrt{\frac{\mu}{3}}(\mathbf{S}^{2n-1,\alpha} - i\mathbf{S}^{2n,\alpha}),$$
(2.35)

where $n = 1, \cdots, 4$ and $\mathbf{Q}_\alpha^I$ and $\mathbf{S}^{I\alpha}$ are the supercharges and the conformal supercharges in OSp(8|4). Plugging this into (B.2), we find the anti-commutator

$$\{Q_\alpha^m, (Q_\beta^n)^\dagger\} = 2\delta_n^m\delta_\alpha^\beta H - \frac{2\mu}{3}\delta_n^m J_\alpha^\beta - \frac{2\mu}{3}\delta_\beta^\alpha R_n^m,$$
(2.36)

where $R_n^m$ are the generators of SU(4) $\subset$ SO(8). This precisely matches with the anti-commutator (2.3) in SU(2|4), with the identification $J_\alpha^\beta = -\frac{1}{2}\epsilon_{ijk}(\sigma^i)_\alpha^\beta M^{jk}$. The Cartan

generators in SU(2|4) and $\mathfrak{osp}(8|4)$ are related by

$$\frac{3}{\mu}H = D - \frac{1}{4}(\mathbf{M}^{12} + \mathbf{M}^{34} + \mathbf{M}^{56} + \mathbf{M}^{78}), \quad M^{12} = J_-^-,$$

$$R_1^1 = \frac{1}{4}(3\mathbf{M}^{12} - \mathbf{M}^{34} - \mathbf{M}^{56} - \mathbf{M}^{78}), \qquad R_2^2 = \frac{1}{4}(-\mathbf{M}^{12} + 3\mathbf{M}^{34} - \mathbf{M}^{56} - \mathbf{M}^{78}),$$

$$R_3^3 = \frac{1}{4}(-\mathbf{M}^{12} - \mathbf{M}^{34} + 3\mathbf{M}^{56} - \mathbf{M}^{78}), \qquad R_4^4 = \frac{1}{4}(-\mathbf{M}^{12} - \mathbf{M}^{34} - \mathbf{M}^{56} + 3\mathbf{M}^{78}),$$

(2.37)

where $\mathbf{M}^{IJ}$ are the SO(8) generators. The Cartan generaotrs $M^{45}$, $M^{67}$, $M^{89}$ of SO(6) $\cong$ SU(4) are related to the Cartan generators of SO(8) by

$$M^{45} = \frac{1}{2}(\mathbf{M}^{12} - \mathbf{M}^{34} - \mathbf{M}^{56} + \mathbf{M}^{78}),$$

$$M^{67} = \frac{1}{2}(-\mathbf{M}^{12} + \mathbf{M}^{34} - \mathbf{M}^{56} + \mathbf{M}^{78}), \qquad (2.38)$$

$$M^{89} = \frac{1}{2}(-\mathbf{M}^{12} - \mathbf{M}^{34} + \mathbf{M}^{56} + \mathbf{M}^{78}).$$

Substituting the relations (2.37) and (2.38) into the formula (2.8) of the Witten index, we find the following relation between the fugacities in (2.8) and (B.6),

$$\gamma_1 = \frac{\Delta_1 - \Delta_2 - \Delta_3}{2}, \quad \gamma_2 = \frac{-\Delta_1 + \Delta_2 - \Delta_3}{2},$$

$$\gamma_3 = \frac{-\Delta_1 - \Delta_2 + \Delta_3}{2}, \quad \gamma_4 = \frac{\Delta_1 + \Delta_2 + \Delta_3}{2}. \qquad (2.39)$$

**Matching the indices in the infinite $N$ limit**

Now, we are ready to compare the BMN index and the ABJM index. The $U(1)_1 \times U(1)_{-1}$ ABJM index is given by the integral,

$$\mathcal{I}_{1,1}^{\text{ABJM}} = \sum_{n,\tilde{n}\in\mathbb{Z}} y_3^{-\frac{1}{2}n} x^{|n-\tilde{n}|} \int_{-\pi}^{\pi} \frac{d\alpha d\tilde{\alpha}}{(2\pi)^2} e^{i(n\alpha - \tilde{n}\tilde{\alpha})}$$

$$\times \exp\left[\sum_{s=\pm}\sum_{m=1}^{\infty} \frac{1}{m} x^{m|n-\tilde{n}|} f_s(x^m, y_1^m, y_2^m) e^{-smi(\alpha-\tilde{\alpha})}\right], \qquad (2.40)$$

where the functions $f_\pm(x, y_1, y_2)$ are given in (B.11). The fugacities $x$, $y_1$, $y_2$, $y_3$ are defined in (B.9). The integral and summation can be easily performed after the change of the integration variables,

$$\alpha_- = \alpha - \tilde{\alpha}, \quad \alpha_+ = \frac{\alpha + \tilde{\alpha}}{2}. \qquad (2.41)$$

The result is

$$\mathcal{I}_{1,1}^{\text{ABJM}} = \exp\left[ \sum_{m=1}^{\infty} \frac{1}{m} \frac{e^{-m\gamma_4}}{e^{-2m\gamma_4}-1} \sum_{\substack{s_i=\pm \\ s_1 s_2 s_3 s_4=-1}} s_4 e^{-\frac{m}{2}(s_1\gamma_1 + s_2\gamma_2 + s_3\gamma_3 + s_4\gamma_4)} \right], \qquad (2.42)$$

Substituting the relation (2.39) between the fugacities into the above formula, we find

$$\mathcal{I}_{1,1}^{\text{ABJM}} = \exp\left\{ \sum_{m=1}^{\infty} \frac{1}{m} \left[ 2 - \frac{(1-e^{-m\Delta_1})(1-e^{-m\Delta_2})(1-e^{-m\Delta_3})}{1-e^{-m(\Delta_1+\Delta_2+\Delta_3)}} \right] \right\}. \qquad (2.43)$$

which agrees with the BMN index (2.34) in the $N \to \infty$ limit up to an divergent factor.

The divergent in the ABJM index is due to the specialization of the chemical potentials (2.39) from four $\gamma_i$'s to three $\Delta_i$'s. A potential explanation of the divergent factor in the relation between the BMN and ABJM indices is that the M2-brane theory on the sphere has infinitely many degenerate vacua that are not distinguished by the three chemical potentials, $\Delta_i$. However, the BMN quantum mechanics only includes one of these vacua that is chosen by the value of $N$. Hence, in this sense the infinity is due to the sum over $N$, which in the ABJM theory would correspond to tracing over different states with the corresponding chemical potential being set to zero.[3]

The full ABJM index, with all four chemical potentials $\gamma_i$ turned on, could potentially be recovered by performing a sum over the BMN indices with different $N$ weighted by an additional chemical potential. We leave this for future work.

# 3  Discussion

In this paper, we compute the Witten index for the BMN matrix quantum mechanics, which counts (with signs) the number of the BPS states. The computation relies on the non-renormalization of the Witten index and is performed at weak coupling, where the theory is divided into superselection sectors associated with each supersymmetric vacuum. The result is a sum of contributions from all superselection sectors, with each term being a matrix integral. In the trivial vacuum sector, we evaluate the matrix integral, and show that it contributes an $N^2$ growth to the entropy. In the irreducible vacuum sector, we find a novel relation between the BMN index and the ABJM index.

Let us discuss some open problems/future directions:

- The $N^2$ growth of the entropy in the trivial vacuum sector indicates that there should exist BPS black holes in M-theory, asymptotic to the plane wave geometry. However,

---

[3]I think Juan Maldacena for suggesting this explanation.

no solution of this sort in the eleven-dimensional supergravity is currently known. Different BPS black hole geometries, with different horizon topologies, could potentially dominate in different regions of the chemical potentials. It is important to extend the saddle point analysis of the Witten index in the trivial vacuum sector to finite $\epsilon_i$ and to other vacuum sectors, as it could shed light on the phase structure of these BPS black holes. The phase structure of non-BPS black holes at finite temperature were explored in [30–36] on the matrix quantum mechanics side by lattice simulations, and [11] on the gravity side.

- The relation between the BMN and ABJM indices can be extended to the case of multiple M2-branes. On the BMN side, this involves the vacua corresponding to direct sums of multiple copies $N$-dimensional irreducible SU(2) representations. Understanding this generalization may require finding a precise relation between the configurations of M2-branes in these two setups, in particular, the relation between the $N$ of the BMN quantum mechanics and a U(1) charge in the ABJM theory.

- Supersymmetric localization of the BMN matrix quantum mechanics was developed in [37], and used in the study of the 1/4-BPS sector [38, 39] and the transverse M5-branes [40, 41]. The same techniques may be used to obtain a path integral derivation of the matrix integral formulae (2.19) with (2.18) for the Witten index.

- Following [14, 42, 43] and more recently [44, 45, 23, 46, 24, 47], we could study the $Q$-cohomology of the BMN matrix quantum mechanics.[4] The $Q$-cohomology classes correspond one-to-one with the BPS states, and the Euler characteristic of the $Q$-cohomology equals the Witten index. The $Q$-cohomology can be straightforwardly computed at weak coupling, although there is no proof or argument that the $Q$-cohomology for BMN matrix quantum mechanics remains invariant when varying the coupling constant $g$.

# Acknowledgments

I am grateful to Sunjin Choi, Shota Komatsu, Ying-Hsuan Lin, and Jorge E. Santos for inspiring discussions, and especially to Juan Maldacena for introducing me to this problem and generously sharing many of his important insights. CC is partly supported by the National Key R&D Program of China (NO. 2020YFA0713000). This research was supported in part by grant NSF PHY-2309135 to the Kavli Institute for Theoretical Physics (KITP). We

---

[4]The supercharge $Q$ is defined above (2.4).

thank the hospitality of the KITP Program "What is String Theory? Weaving Perspectives Together" and Conference "Spacetime and String Theory".

# A  Small black holes in AdS$_5$

In this appendix, we review the discussions about small AdS black holes in [26]. The BPS black hole in AdS$_5$ has a partition function given by the Euclidean gravity path integral as

$$\log Z = \frac{N^2}{2} \frac{\Delta_1 \Delta_2 \Delta_3}{\omega_1 \omega_2}, \quad \Delta_1 + \Delta_2 + \Delta_3 - \omega_1 - \omega_2 = 2\pi i. \tag{A.1}$$

Let us consider the specialization to only one fugacity $x = e^{-\beta}$ as

$$\Delta_i = 2\beta + 2\pi i, \quad \omega_i = 3\beta + 2\pi i. \tag{A.2}$$

Since the thermal AdS phase has a free energy of order 1, the black hole phase is in the region

$$\mathrm{Re}\,(\log Z) = \mathrm{Re}\,\left(\frac{N^2}{2} \frac{(2\beta + 2\pi i)^3}{(3\beta + 2\pi i)^2}\right) > 0. \tag{A.3}$$

We plot this region in blue on the $|x|$-$\phi$ plane in Figure 2, where $x = e^{-\beta} = |x|e^{i\phi}$.

The large/small black holes are in the region of positive/negative specific heat $C$ (susceptibility) given by

$$C = \mathrm{Re}\,\left(\beta_{\mathrm{R}}^2 \frac{d^2 \log Z}{d\beta_{\mathrm{R}}^2}\right), \tag{A.4}$$

where $\beta_{\mathrm{R}} = \mathrm{Re}\,\beta = -\log|x|$. We plot the region of positive specific heat in orange in Figure 2.

The entropy of the black hole is given by extremizing the function

$$\begin{aligned}
S(\beta; Q, J) &= \log Z + Q_1 \Delta_1 + Q_2 \Delta_2 + Q_3 \Delta_3 + J_1 \omega_1 + J_2 \omega_2 \big|_{Q_i = Q,\, J_i = J,\, (\mathrm{A.2})} \\
&= \frac{N^2}{2} \frac{(2\beta + 2\pi i)^3}{(3\beta + 2\pi i)^2} + 3Q(2\beta + 2\pi i) + 2J(3\beta + 2\pi i).
\end{aligned} \tag{A.5}$$

This function has three extrema. We pick the one that gives a positive real part of $S$. Imposing the condition that the imaginary part of $S$ vanishes, we find

$$\mathrm{Im}\,S = 0 \quad \Rightarrow \quad Q^3 + \frac{N^2}{2} J^2 = \left(\frac{N^2}{2} + 3Q\right)(3Q^2 - N^2 J), \tag{A.6}$$

which admits one solution with $Q, J > 0$. The entropy at the chosen extremum is

$$S(Q) = 2\pi \sqrt{3Q^2 - N^2 J}, \tag{A.7}$$

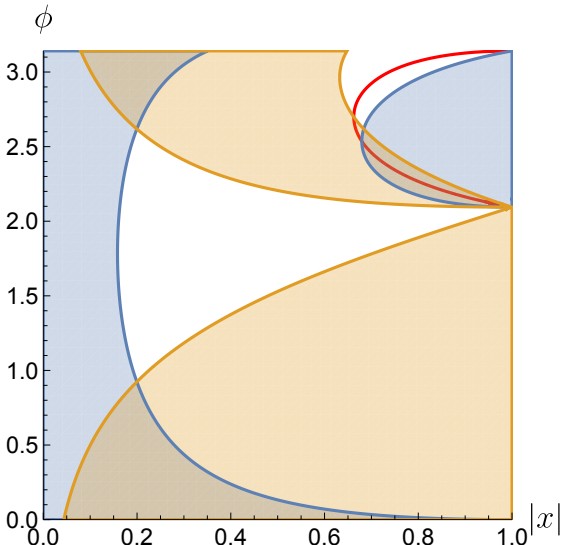

Figure 2: The blue and orange regions are given by (A.3) and $C > 0$ for $C$ given in (A.4). The red curve is given by the function $\beta(Q)$ for $Q$ varying from 0 to $\infty$.

which is always positive for $Q, J > 0$ satisfying the condition (A.6).

Let us write the chosen extremum as $\beta(Q)$, after substituting $J$ using (A.6). We plot $\beta(Q)$ on the $|x|$-$\phi$ plane as the red curve in Figure 2 for $Q$ varing from 0 to $\infty$. The small charge limit $Q \to 0$ is at $(|x|, \phi) = (1, \pi)$, and the large charge limit $Q \to \infty$ is at $(|x|, \phi) = (1, \frac{2\pi}{3})$. There is an intersection between the curve $\beta(Q)$ and the phase boundary at

$$Q_* = \frac{N^2}{24}(1 + \sqrt{33}).$$
(A.8)

We see that when increasing $Q$ from 0 to $\infty$, the red curve is first inside the white region ($\log Z < 0$ and $c < 0$) corresponding to a small black hole phase that is subdominant to thermal AdS in the grand canonical ensemble. Then, the red curve enters the orange region ($\log Z < 0$ and $c > 0$) corresponding to a large black hole phase that is also subdominant. Finally, the red curve enters the orange and blue region ($\log Z > 0$ and $c > 0$) corresponding to a large and dominant black hole phase. While the small black holes are always subdominant, they still carry macroscopic entropy (A.7).

# B    Review of ABJM index

## B.1    $\mathcal{N} = 8$ superconformal index

The $\mathcal{N} = 8$ ABJM theory has superconformal symmetry OSp(8|4). Let us denote the supercharges and the conformal supercharges in OSp(8|4) by

$$\mathbf{Q}_\alpha^I, \quad \mathbf{S}^{I\alpha}, \tag{B.1}$$

where $I = 1, \cdots, 8$ is the vector index of SO(8) and $\alpha = \pm$ is the spinor index of SU(2) $\subset$ Sp(4). We have the anti-commutator

$$\{\mathbf{Q}_\alpha^I, \mathbf{S}^{J\beta}\} = \delta^{IJ}\delta_\alpha^\beta D - \delta^{IJ}J_\beta^\alpha - i\delta_\beta^\alpha \mathbf{M}^{IJ}, \tag{B.2}$$

where $D$ is the dilatation charge, $J_\beta^\alpha$ are the generators of SU(2), and $\mathbf{M}^{IJ}$ are the generators of SO(8). Let us define

$$\boldsymbol{\Delta} := \frac{1}{2}\{\mathbf{Q}_-^7 + i\mathbf{Q}_-^8, \mathbf{S}^{7-} - i\mathbf{S}^{8-}\} = D - J - \mathbf{M}^{78}, \tag{B.3}$$

where $J := J_-^-$. Let us consider the partition function

$$Z = \mathrm{Tr}\,\Omega, \quad \Omega := e^{-\beta\boldsymbol{\Delta}-2\omega J - \gamma_1\mathbf{M}^{12} - \gamma_2\mathbf{M}^{34} - \gamma_3\mathbf{M}^{56} - \gamma_4\mathbf{M}^{78}}. \tag{B.4}$$

The supercharge $\mathbf{Q}_-^7 + i\mathbf{Q}_-^8$ anti-commutes with the Boltzmann factor $\Omega$ if

$$\gamma_4 - \omega = 2\pi i \mod 4\pi i. \tag{B.5}$$

We define the superconformal index as

$$\mathcal{I} = Z\big|_{\text{(B.5)}} = \mathrm{Tr}\left[(-1)^F e^{-\beta\boldsymbol{\Delta} - \gamma_1\mathbf{M}^{12} - \gamma_2\mathbf{M}^{34} - \gamma_3\mathbf{M}^{56} - \gamma_4(2J + \mathbf{M}^{78})}\right], \tag{B.6}$$

where we identify $2J$ with the fermion number $F$, since all the fermionic states have half-integer spin $J$. By the standard arguments, the superconformal index $\mathcal{I}$ is independent of the inverse temperature $\beta$.

## B.2   Integral formula

The superconformal index for the $\mathrm{U}(N)_k \times \mathrm{U}(N)_{-k}$ ABJM theory was computed in [18],[56]

$$
\begin{aligned}
\mathcal{I}_{N,k}^{\mathrm{ABJM}} = \frac{1}{(N!)^2} \sum_{n_i, \tilde{n}_i \in \mathbb{Z}} y_3^{-\frac{k}{2} \sum_{i=1}^N n_i} \int_{-\pi}^{\pi} \prod_{i=1}^N \frac{d\alpha_i d\tilde{\alpha}_i}{(2\pi)^2} e^{ik(n_i\alpha_i - \tilde{n}_i\tilde{\alpha}_i)} \\
\times \exp\left[ -\omega\epsilon_0 + \sum_{m=1}^{\infty} \frac{1}{m} f(x^m, y_1^m, y_2^m, e^{im\alpha_i}, e^{im\tilde{\alpha}_i}) \right],
\end{aligned}
\tag{B.8}
$$

where the fugacities $x$ and $y_i$ are

$$
x = e^{-\gamma_4}, \quad y_1 = e^{-\gamma_1}, \quad y_2 = e^{-\gamma_2}, \quad y_3 = e^{-\gamma_3}.
\tag{B.9}
$$

The $\epsilon_0$ and $f$ are

$$
\epsilon_0 = \sum_{i,j} |n_i - \tilde{n}_j| - \sum_{i<j} (|n_i - n_j| + |\tilde{n}_i - \tilde{n}_j|),
\tag{B.10}
$$

and

$$
\begin{aligned}
f(x, y_1, y_2, e^{i\alpha_i}, e^{i\tilde{\alpha}_i}) &= -\sum_{i \neq j} x^{|n_i - n_j|} e^{-i(\alpha_i - \alpha_j)} + \sum_{i,j} f_+(x, y_1, y_2) x^{|n_i - \tilde{n}_j|} e^{-i(\alpha_i - \tilde{\alpha}_j)} \\
&\quad - \sum_{i \neq j} x^{|\tilde{n}_i - \tilde{n}_j|} e^{-i(\tilde{\alpha}_i - \tilde{\alpha}_j)} + \sum_{i,j} f_-(x, y_1, y_2) x^{|\tilde{n}_i - n_j|} e^{-i(\tilde{\alpha}_i - \alpha_j)}, \\
f_+(x, y_1, y_2) &= \frac{1}{1-x^2} \left[ x^{\frac{1}{2}} \left( \sqrt{\frac{y_1}{y_2}} + \sqrt{\frac{y_2}{y_1}} \right) - x^{\frac{3}{2}} \left( \sqrt{y_1 y_2} + \frac{1}{\sqrt{y_1 y_2}} \right) \right], \\
f_-(x, y_1, y_2) &= f_+(x, y_1, y_2^{-1}).
\end{aligned}
\tag{B.11}
$$

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
