# Peer review of "Witten index of BMN matrix quantum mechanics"

_SciPost Physics_

## Round 1 · Referee Report · Anonymous (Referee 1) · 2025-8-11

Report
This paper computes the Witten index of the BMN matrix quantum mechanics in the trivial and irreducible vacuum sectors. The results shed light on possible dual black hole configurations and the fuzzy sphere interpretation of the different vacua. They are timely, connect with several active lines of research, and open promising directions for future work.
The results are interesting and relevant, but there are several points in the presentation that would benefit from improved clarity and further explanation:
1) In computing the Witten index, the BMN matrix quantum mechanics is analytically continued from Lorentzian to Euclidean signature. This continuation can either: - change the R-symmetry to $SO(1,2)\times SO(6)$ or $SO(3)\times SO(1,5)$, or - change the signature of the original eleven-dimensional space to be Euclidean. In this case the R-symmetry remains $SO(3) \times SO(6)$ but the spinors (and hence the supercharges) must be complexified. While these choices should not affect the final result, they can alter the space of classical vacua. It would be helpful for the author to briefly discuss these subtleties and clearly state which perspective is adopted.
2) The discussion of the perturbative spectrum in each vacuum sector is somewhat difficult to follow and appears to contain inconsistencies with Ref. [13]. The author adopts a different notation, $N = \sum n_k N_k$ instead of $N= \sum_k N_k$, which is perfectly acceptable. However, with this choice, expressions such as (2.16) should be updated accordingly for consistency.
3) In equations (2.17)-(2.18), the notation $z_{kl}$, $i_{kl}$ is somewhat confusing. In the conventions of this paper, it appears that a summation over the multiplicity of the relevant su(2) representation is missing. In (2.19), this seems to be compensated by the presence of the U_k matrices. Clarifying this point would improve readability.
4) $N^2$ growth of the index is a necessary but not sufficient condition for the existence of black holes. The author should clarify their position on this point: do they claim that black holes exist, or that the results provide evidence for their existence? Furthermore, it should be specified whether the putative black holes are large or small. The current discussion suggests similarities with small black holes, but these typically yield a slower growth of the index.
5) There are numerous typographical errors, including missing articles. A careful spelling and grammar check would substantially improve the readability of the paper.
Provided the points above are adequately addressed, I would recommend this manuscript for publication.
Requested changes
See list above.
Recommendation
Ask for minor revision
We thank the referee for the report. 1. In the paper, the Witten index is computed using the Hamiltonian formalism. We added footnote 1, which comments on the computation in the path integral formalism that relies on the analytic continuation from Lorentzian to Euclidean signature. 2. We improved the paragraph containing Equations (2.15) - (2.17). Equation (2.15) does not need to be updated. We just need to regard $x^a_{kl,jm}$ as an $n_k\times n_l$ matrix. 3. We improved the paragraph on page 7. We added (2.18), which includes the bi-fundamental character, and renamed $z_{kl}$, $\iota_{kl}$ as the colorless single-letter BPS partition function and index (respectively). 4. We thank the referee for emphasizing this point. We changed wording to “provides strong evidence”, “strongly suggests”, and “may correspond to”. We added footnote 4 to emphasize the properties of the small black holes in AdS5. 5. We corrected all the spelling and grammar that we can find.

Author: Chi-Ming Chang on 2025-10-07 [id 5897]
(in reply to Report 2 on 2025-08-18)We thank the referee for the report. 1. We improved paragraph 3 on page 2. 2. We added (2.48) to clarify the meaning of the divergent factor. Unfortunately, we do not have anything more to add about the resolution of this discrepancy and the further matching beyond the irreducible vacuum sector. This is why we leave this for future work.

---

## Round 1 · Referee Report · Anonymous (Referee 2) · 2025-8-18

Report
In this paper the author computes the Witten index of BMN matrix quantum mechanics. It is found that the Witten index grows as $N^2$ which is consistent with BPS black holes. An interesting relation between the BMN index and the index of U(1) ABJM theory is discussed. These are interesting and relevant results that deserve publication in Scipost. I have a couple of minor comments on the manuscript:
- The discussion on page 2 paragraph 2 introduces the Witten index in quantum mechanics in fairly general terms. In particular the Witten index is related to the number of ground states. Then in paragraph 3 the author moves on to discussing the computation of the Witten index for the BMN matrix quantum mechanics which can be split into computing the Witten index in each vacuum sector. This may be confusing to unfamiliar readers when compared to paragraph 2 and so further explanations may be added.
- In section 2.5, the author finds a relation between the index restricted to the irreducible vacuum and the U(1) ABJM index. However the match is only discovered up to a divergent factor. The difference is then argued to stem from a lack of a quantum number in the BMN description that is there in ABJM. This is then further related to N which is there in the BMN description and not the ABJM description which possibly may be related to the extra quantum number (If I understood the 2nd paragraph on page 12 correctly). I think this interesting point could be expanded upon. What exactly is the divergent factor discrepancy? (is the factor 2 in (2.43) supposed to be 1 as in (2.34)?) Is the claim that going to the grand canonical ensemble for (2.34) will yield back (2.43)? The author mentions also the relation of reducible vacua to ABJM with higher gauge groups, but what would the full index (2.20) correspond to in ABJM?
Recommendation
Ask for minor revision

---

## Editorial Decision

unknown